# An Explainable Deep Learning Classifier of Bovine Mastitis Based on Whole-Genome Sequence Data—Circumventing the p >> n Problem

**DOI:** 10.3390/ijms25094715

**Published:** 2024-04-26

**Authors:** Krzysztof Kotlarz, Magda Mielczarek, Przemysław Biecek, Katarzyna Wojdak-Maksymiec, Tomasz Suchocki, Piotr Topolski, Wojciech Jagusiak, Joanna Szyda

**Affiliations:** 1Biostatistics Group, Department of Genetics, Wroclaw University of Environmental and Life Sciences, Kozuchowska 7, 51-631 Wroclaw, Poland; krzysztof.kotlarz@upwr.edu.pl (K.K.); magda.mielczarek@upwr.edu.pl (M.M.); tomasz.suchocki@upwr.edu.pl (T.S.); 2University Cancer Diagnostic Center, Poznan University of Medical Science, 61-701 Poznan, Poland; 3Faculty of Mathematics, Informatics and Mechanics, University of Warsaw, Banacha 2, 02-097 Warsaw, Poland; przemyslaw.biecek@pw.edu.pl; 4Faculty of Mathematics and Information Science, Warsaw University of Technology, 00-662 Warsaw, Poland; 5Department of Genetics and Animal Breeding, West Pomeranian University of Technology, Aleja Piastow 45, 70-311 Szczecin, Poland; katarzyna.wojdak-maksymiec@zut.edu.pl; 6National Research Institute of Animal Production, Krakowska 1, 32-083 Balice, Poland; piotr.topolski@izoo.krakow.pl (P.T.); wojciech.jagusiak@urk.edu.pl (W.J.); 7Faculty of Animal Science, University of Agriculture in Krakow, al. Mickiewicza 24/28, 30-059 Kraków, Poland

**Keywords:** artificial intelligence, cattle, clinical mastitis, deep learning, enrichment, Holstein–Friesian, SNP

## Abstract

The serious drawback underlying the biological annotation of whole-genome sequence data is the p >> n problem, which means that the number of polymorphic variants (p) is much larger than the number of available phenotypic records (n). We propose a way to circumvent the problem by combining a LASSO logistic regression with deep learning to classify cows as susceptible or resistant to mastitis, based on single nucleotide polymorphism (SNP) genotypes. Among several architectures, the one with 204,642 SNPs was selected as the best. This architecture was composed of two layers with, respectively, 7 and 46 units per layer implementing respective drop-out rates of 0.210 and 0.358. The classification of the test data resulted in AUC = 0.750, accuracy = 0.650, sensitivity = 0.600, and specificity = 0.700. Significant SNPs were selected based on the SHapley Additive exPlanation (SHAP). As a final result, one GO term related to the biological process and thirteen GO terms related to molecular function were significantly enriched in the gene set that corresponded to the significant SNPs. Our findings revealed that the optimal approach can correctly predict susceptibility or resistance status for approximately 65% of cows. Genes marked by the most significant SNPs are related to the immune response and protein synthesis.

## 1. Introduction

Due to the development of high-throughput technology, the past few decades have seen a considerable increase in the availability of genomic data [1,2]. Among them, the most common data structure is the whole-genome sequence (WGS) that is nowadays available for thousands of individuals representing various species, e.g., the European 1+ Million Genomes Initiative for humans (https://digital-strategy.ec.europa.eu/en/policies/1-million-genomes) or the 1000 Bull Genomes Project for cattle [3]. Effective and efficient computing methods are emerging issues regarding the storage, analysis, and interpretation of this flood of biological information [4,5]. However, the most serious drawback underlying the utilization of WGS data is their statistical nature, the so-called p >> n problem, which means that the number of predictors, i.e., polymorphic variants (p) is much larger than the number of available phenotypic records (n) [6]. This impedes the application of standard statistical models, such as regression, unless we decide to split the available predictors into a single (oligo) predictor analysis, which is often the case in Genome-Wide Association Studies (GWAS), in which, despite the availability of millions of polymorphic variants to test their association with phenotypes, many single-variant models are applied [7,8], followed by multiple testing correction of the individual hypothesis tests. However, this is related to the loss of an important source of information contained in high-throughput data, that is, the interaction between particular predictors. One possible way around the p >> n problem is to use models that impose some shrinkage in the estimation of effects—like, e.g., mixed models, ridge regression, or LASSO. Another recent trend is to use deep learning (DL), which in many areas offers higher accuracy in classification or prediction. DL has been increasingly used in computational biology, for example, in genomics to identify regulatory variants [9] or in clinical genetics to predict the effect of mutations [10] applied to a wide range of biological materials, from single cells [11] to tissues [12]. However, despite the great flexibility regarding analyzed data structures, a critical problem in using DL is the underlying complexity of the algorithms, which makes it difficult to interpret the outcome in terms of formally defined statistical hypotheses and, consequently, to formulate biologically interpretable conclusions.

Bovine mastitis is a disease that is one of the most common disorders in dairy cows [13,14,15], causing problems in animal welfare and economic losses. Mastitis accounts for 38% of all direct costs associated with major production disorders, as well as 70% of the total losses attributed to mammary tissue injury. This reduces milk production [16] and therefore remains the most economically significant disease that affects dairy cattle [17]. The occurrence of bovine mastitis is known to be significantly influenced by several risk factors, including pathogens, host genetics, and the environment.

Therefore, the main objective of this study was to propose a way to circumvent the p >> n problem by combining a LASSO logistic regression model and DL illustrated by a practical biological problem of classifying cows into mastitis-susceptible or mastitis-resistant, based on genotypes of Single Nucleotide Polymorphisms (SNPs) identified in their whole-genome DNA sequences. This translates into the situation that the number of available SNPs (p) vastly exceeds the number of analyzed cows (n). Furthermore, we tackle the problem of biological explainability of results at a single SNP level using SHapley Additive exPlanation values (SHAP) [18]. Biologically, our aim was to identify the biological processes, molecular functions, cellular components, and metabolic pathways that are the most affected by the incidence of clinical mastitis.

## 2. Results

### 2.1. Data Processing

Due to the poor genome averaged coverage of 3X resulting after the alignment to the reference genome, one cow was removed from the training group. The average genome coverage for the remaining 31 individuals in the training group ranged between 7X and 13X, while for the 20 individuals in the test group, it varied between 14 and 37X. After filtering, 16,618,983 SNPs were considered in the downstream analysis. Since the SNP call was performed separately for testing and training individuals and SNP genotypes that were polymorphic in only one of the data sets, the other group was set as a homozygous reference, which is the most frequent genotype constellation.

### 2.2. The Optimal DL Architecture

The number of SNPs preselected using the penalized regression approach varied between 6665 for the highest penalty expressed by C = 0.1 and 1,154,608 for the mildest penalty expressed by C = 1.0. Figure 1 presents the number of SNPs selected along the decreasing penalty with each subsequent set of SNPs containing variants from the preceding subsets, $\left(C_{n-1\;}\subseteq C_{n}\right)$. A markedly different DL architecture was selected as the optimal one, depending on the SNP set. The number of layers ranged from one (for C = 0.1 and 0.9) to four (C = 0.8). For none of the subsets, the maximum allowed number of layers was estimated as optimal. The number of units per layer varied between seven and 50, and the dropout rates ranged from 0.215 to 0.398 (Table 1).

### 2.3. The Classification Quality

The classification quality expressed by the AUC for each of the estimated DL architectures calculated based on the 4-fold cross-validation of the training data is summarized in Figure 2A, which shows that with AUCs varying between 0.925 (C = 0.1) and 1.000 (C = 0.9), all algorithms provided a reasonable classification. Furthermore, the validation loss generally decreased with an increasing number of SNPs included in the model, varying between 0.925 for the most parsimonious model with C = 0.1 and 0.287 for the model with C = 0.9. However, when applied to the test data, the classification quality decreased considerably and varied between 0.400 for the SNP set selected under C = 0.6 and 0.750 for C = 0.4 (Figure 2B). The highest loss in AUC by 0.504 was observed for the SNP set defined by C = 0.6, while the subsets selected with C = 0.1, C = 0.4, and C = 0.5 were the most robust with 0.252, 0.177, and 0.256 decrease in AUC, respectively.

### 2.4. Selection of the Best DL Architecture

For all SNP sets considered, the estimated classification cut-off values differed from the default of 0.5 (Figure 3). However, they also differed considerably from each other, from 0.142 (C = 0.9) to 0.893 (C = 0.2). The cut-off values estimated based on the more complex DL architectures (0.2, 0.3, and 0.6) were higher than those estimated based on parsimonious architectures. Therefore, the most parsimonious DL architecture with only one layer underlying the SNP set obtained under C = 0.8 resulted in a very low cut-off value of 0.142. For each DL architecture, the application of the optimal cut-off value for the classification of the training data resulted in a higher ACC than using the default cut-off value of 0.5 (Figure 4A). The greatest improvement of 0.419 was reached for C = 0.4, an SNP set based on the default threshold did not even reach the accuracy of a random group assignment (i.e., 0.500). The architecture underlying C = 0.9 resulted in a “perfect” accuracy of one, even for the default cut-off, indicating model overfitting. However, when applied to the test data sets, the estimated cut-off values did not always result in a better classification. The strongest increase in accuracy by 0.200 was obtained for a SNP set selected based on C = 0.7 (Figure 4B). Furthermore, the classification accuracy of the test data was much lower compared to the training data sets and oscillated around 0.500. The highest overall test accuracy of 0.700 was achieved for the data set generated under C = 0.9 and the default cut-off value. Three SNP sets (C = 0.1, C = 0.4, and C = 0.7) also resulted in a reasonable accuracy of 0.650 by using estimated cut-off values. The standard errors of the cut-off points did not exceed 0.050, indicating high accuracy of the cut-off values (Figure 3). Figure 4A,B visualize the ACC classification differences obtained for the training and test data set with the default cut-off and the optimal cut-off of the classification algorithms.

Another important classification metric from the practical perspective is sensitivity (Figure 5A), which reflects the algorithm’s ability to correctly classify an animal as susceptible to mastitis. Although a very high sensitivity, ranging from 0.750 (C = 1.0) to 1.000 (C = 0.1, C = 0.3, and C = 0.9) was reached for the training data sets, the classification sensitivity of the test data was very low, except for a classification model for C = 0.1, which obtained a high sensitivity of 0.800. On the other hand, the specificity (Figure 5B) of the classification of the test data classification (i.e., the ability to correctly classify an animal as mastitis-resistant) was generally higher than the sensitivity. With that, it became evident that for most algorithms, the identification of resistant individuals is easier. The classification of the test data also reveals an interplay between both metrics; consequently, models with high sensitivity result in low specificity and the opposite. As an additional performance metric, the Matthews correlation coefficient (MCC) (Figure 5C) was calculated for the training and test data sets, respectively. A comparison of performance metrics for all models with different C parameters and thresholds, including AUC from LASSO logistic regression models, is given in Appendix A.

By summarizing the performance of the different DL architectures expressed by AUC, accuracy, sensitivity, and specificity, the model corresponding to C = 0.4, characterized by the test AUC of 0.750, ACC equal to 0.650 (for the optimal cut-off), the appropriate value of SENS = 0.600 and the high value of SPEC = 0.700 were selected as the best model and, thus, proceeded to genomic and functional annotations. This architecture uses 204,642 SNPs.

### 2.5. Genomic and Functional Annotation

Of the 204,642 SNPs that comprise the best model, 3162 obtained significant (α ≤ 0.05) SHAP values (Figure 6), and 1235 of those SNPs could be annotated to 966 genes—by being located within their coding sequence (23 synonymous and 18 missense SNPs). Within non-coding sequences, significant positions were located in introns (856 SNPs), non-coding transcripts (33 SNPs), non-coding exon variants (4 SNPs), upstream of genes (180 SNPs), downstream of genes (101 SNPs), 3′UTR variants (7 SNPs), or 5′UTR variants (13 SNPs). The remaining positions (1927 SNPs) were intergenic variants, and 746 of these SNPs were novel. The 966 genes were then tested for enrichment in Gene Ontology (GO) terms from the biological process and the molecular function category, as well as in the KEGG and Reactome pathways. As a result, a single GO term related to biological processes (GO:0050804~modulation of chemical synaptic transmission) and 13 GO terms related to molecular function (GO:0005509~calcium ion binding, GO:0030554~adenyl nucleotide binding, GO:0032559~adenyl ribonucleotide binding, GO:0005524~ATP binding, GO:0032553~ribonucleotide binding, GO:0032555~purine ribonucleotide binding, GO:0017076~purine nucleotide binding, GO:0005216~ion channel activity, GO:0031267~small GTPase binding, GO:0005096~GTPase activator activity, GO:0017075~syntaxin-1 binding, GO:0022838~substrate-specific channel activity, GO:0008276~protein methyltransferase activity) were significantly enriched, but none of the GO terms described cellular components, nor KEGG or the Reactome pathway (Appendix A). The assignment of SNP to the significant gene ontologies is summarized in Appendix A.

### 2.6. GWAS for Clinical Mastitis in Polish Holstein–Friesian Cows

In the association study of the large data set, 3188 SNPs were significantly associated with clinical mastitis. Most of these SNPs (99.72%) successively remapped to ARS-UCD1.2 and annotated 1209 genes, revealing 184 genes in common with genes marked by significant SNPs from the C = 0.4 set. To further compare the functional annotation resulting from GWAS with that of the DL model, genes marked by significant GWAS SNPs were tested for enrichment of GO terms. As a result, 11 GO terms from biological processes, 28 GO terms from molecular function, and 8 GO terms from cellular components were significantly enriched. Eight GO terms (GO:0005509~calcium ion binding, GO:0030554~adenyl nucleotide binding, GO:0032559~adenyl ribonucleotide binding, GO:0005524~ATP binding, GO:0032553~ribonucleotide binding, GO:0032555~purine ribonucleotide binding, GO:0017076~purine nucleotide binding, GO:0005216~ion channel activity, GO:0005096~GTPase activator activity, GO:0022838~substrate-specific channel activity) overlap between significant enrichment based on GWAS.

## 3. Discussion

Susceptibility to bovine mastitis is a complex trait since it is determined by a wide variety of bacteria, non-biological components (e.g., maintenance of milking equipment or post-milking teat disinfection) as well as by the genetic composition of an individual [19,20]. Depending on the etiology, the heritability of clinical mastitis varies between 0.01 and 0.25 [21], the latter indicating the considerable impact of the genetic component. Moreover, the definition of mastitis varies from sub-clinical mastitis that is manifested mainly by elevated somatic cell count in milk but lacks visible symptoms to clinical mastitis that involves alteration of the udder [22]. In our study, using clinical mastitis as an example, we suggest and test a new three-step pipeline involving bioinformatic, statistical, and biological components to unravel the functional component of a disease underlying a complex mode or inheritance.

### 3.1. Data Processing

In a highly dimensional data set, it is difficult to determine which of the explanatory variables are relevant [23] for prediction or classification. Furthermore, many of the explanatory variables are highly correlated, so they do not provide unique information. Moreover, due to the number of features greater than the number of observations, it is difficult not only to build the model due to overfitting, but also to provide a concise and reproducible interpretation of its results, due to a wide range of significant hits [24]. In the feature selection process, the originally very large number of explanatory variables (SNPs) was reduced to describe the response variable (mastitis-susceptible or mastitis-resistant), using the tuning parameter  $C=\frac{1}{\lambda }$ that regulates the sparsity of the estimator (i.e., the number of zero-valued coefficients), an approach similar to Fallerini et al. [25] that, however, appeared to use a single predefined penalty value.

During this process, two major issues emerged. First, how do we choose the number of SNPs? On the one hand, the lower number of SNPs makes the prediction model less computationally expensive, but on the other hand, it may result in a larger prediction error [26]. The problem can be extrapolated from genomics to database handling, where selectivity estimation is related to estimating the number of records that satisfy query conditions [27]. However, the selectivity approach proposed in our study uses quasi-empirical modeling of the LASSO penalty parameter λ by exploring the classification quality metrics of DL algorithms underlying a predefined range of penalty parameters that cover the full scope of potentially available SNPs. The second issue is the choice of the most appropriate measure of classification quality [28]. The ACC metric, which is the standard in the evaluation of classification quality, may become misleading when classification class sizes are imbalanced. Although in our data both class sizes were almost completely balanced, the drawback still exists that this classification quality metric relies on a binary assignment of individuals to TP/FP/TN/FN groups, regardless of the actual probability of classification, which is the primary output of the sigmoid activation function from the last layer. To mitigate this problem, we proposed exploring the whole range of the probability parameter space (i.e., [0, 1]) and the estimation of the cut-off value that guarantees the best ACC. However, the data resampling approach requires a large sample size, which was not the case in our study. Another proposal is to evaluate the quality of the classification based on AUC that compares multiple thresholds of true-positive (TPR) and false-positive (FPR) rates [29] and provides a metric that simultaneously accounts for the sensitivity and high specificity [30] of the classification with a strong emphasis on the sensitivity to cover individuals susceptible to mastitis, minimizing the chances of false negatives, which is crucial for the efficient cow treatment.

Note that none of the DL architectures compared could be unequivocally classified as the best model by reaching the top scores for all metrics applied. Practically, this means that there will be individuals that are incorrectly labeled as susceptible or resistant. So, the practical element of the best model selection is also driven by the interests of the end user of the classification, like milk producers in the case of our data or, e.g., clinicians in the case of medical data.

### 3.2. Functional Interpretation of Significant SNPs

The Gene Ontology resource is the world’s largest source of information on gene functions. It defines biological domains expressed as molecular functions, biological processes, and cellular components. GO enrichment analysis revealing molecular-level activities performed by gene products was essential for the functional interpretation of significant SNPs. All molecular functions significant in this study are fundamental in nearly every aspect of cell biology. According to Neculai-Valeanu and Ariton [31], mastitis causes alterations in the ionic dynamics of the vascular components and is primarily caused by massive cellular destruction and a weak milk–blood barrier. An increased concentration of ions in milk during mastitis infection involves sodium, potassium, calcium, magnesium, and chloride [32], which explains the overrepresented GOs related to ion binding and activity found in this study. Monoatomic ion channel activity (GO:0005216), according to the AmiGo definition [33], facilitates the diffusion of ions during their passage through a transmembrane aqueous pore or channel. However, calcium ion binding ontology (GO:0005509) has been reported in the context of primary mammary epithelial cells (PMECs) infection, where suppression of this ontological term was caused by Lipopolysaccharide (LPS), a toxin located in the outer membrane of Gram-negative bacteria [34]. Although this molecular function was suppressed, the authors observed that only a few ontological categories suppressed by LPS were significant, and they hypothesized that LPS induces immune, inflammatory, and defense responses. In fact, immunological defense is a very energetically costly process that requires a change in energy from less essential metabolic functions to the immune system in the presence of pathogens, which explains the role of ATP in the inflammation process. The ATP binding GO term (GO:0005524) was previously reported to be the most representative molecular function in the context of *Mycoplasma bovis* infection. This species is one of the main bovine pathogens that cause multiple diseases, including mastitis [35]. Furthermore, due to the ATP substrate mentioned above and its specific channels, alterations in the number of molecules that activate these channels influence their activity (GO:0022838), causing, in the case of bovine mastitis, an increase in the inflammatory response, which also indicates the role of GTP binding in the immune response to bovine mastitis, and which overlaps with our findings of the significant molecular functions of small GTPases binding (GO:0031267) and the activity of the GTPase activator (GO:0005096). Furthermore, GTPase-regulated pathways have often been mentioned in the context of mastitis, especially with respect to inflammation caused by *Streptococcus agalactiae* and *Mycoplasma bovis* [35,36,37,38,39,40]. Inflammation, which is the consequence of infection, involves exocytosis that leads to the release of granule/vesicle contents to the cell exterior. It is of particular importance with respect to tissue damage, being a consequence of inflammatory cell activation and mediator development [41]. Syntaxins are protein families that play an important role in exocytosis and can help explain the molecular process of the binding of syntaxin-1 (GO:0017075), which was found to be significant in this study. Regarding adenyl ribonucleotide binding (GO:0032559), not much has been reported in the mastitis-related literature. However, miRNA expression profiles were investigated in porcine mammary epithelial cells after contact with a potential Escherichia coli strain causing mastitis. The predicted target genes for miRNAs regulated up and down were significantly enriched in molecular functions, including, among others, adenyl ribonucleotides [42]. Furthermore, adenyl nucleotide binding (GO:0030554) and the ribonucleotide binding term (GO:0032553) were reported to be significant in the altered molecular expression of the signaling pathway in mammary tissue from cattle with mastitis. The former was enriched in genes overexpressed in the mammary tissue of cows infected with mastitis [43]. Furthermore, the nervous system plays a role in infection response since the immune system communicates with the nervous system to coordinate the immune response through signaling molecules, such as neurotransmitters and neuromodulators [44], which is reflected in the significant enrichment of the ontology of modulation of chemical synaptic transmission (GO:0050804). Purines are the bases of DNA and RNA that are required for the synthesis of nucleic acids and then proteins and other metabolites, as well as for reactions that require energy [45]. The relation between nucleotide biosynthesis and bacterial pathogenesis in diseases was reported by Goncheva et al. [46] and demonstrated a connection with the purine ribonucleotide and nucleotide binding ontologies (GO:0032555, GO:0017076) significant in our study. Finally, at the epigenetic level, Usman et al. [47] reported that the promoter regions of the JAK2 and STAT5A genes were hypomethylated in cows with mastitis, which is consistent with the significance of the protein methylation ontology (GO:0008276) estimated in our study.

## 4. Materials and Methods

### 4.1. Sequenced Animals

A total of 52 Polish Holstein–Friesian cows from the same herd were selected with 991 clinical cases of mastitis diagnosed by a veterinarian. All cows were kept in the same barn under unified conditions and fed the same balanced diet. Furthermore, they were born in the same year and season and also calved at the same age. The cows were daily pre-examined by the farm staff during the attachment of milking cups in a fishbone milking parlor. Subsequently, all suspected cases of mastitis were reported to the farm’s resident veterinarian, who made the final diagnosis of udder inflammation based on clinical symptoms such as redness and swelling of the teats and udder, presence of blood and/or pus streaks and flecks in the milk, elevated temperature, and tenderness of the udder. The 52 individuals were divided into a training group of 32 cows and a test group of the remaining 20 cows. In the training group, the cows were paternal half-sibs matched by the number of recorded parities, production level, and birth year, but differed in their resistance status to mastitis. So, 16 cows were mastitis-resistant and had no incidence of clinical mastitis throughout their production life, while 16 mastitis-susceptible cows underwent multiple disease incidences. Their genomic DNA was sequenced with the Illumina HiSeq2000 platform in paired-end mode with a read length of 100 bp. The number of raw reads generated for a single animal ranged from 164,984,147 to 472,265,620. The experimental design and the training data set were described in detail by Szyda et al. [48]. The test group consisted of 10 mastitis-susceptible cows and 10 mastitis-resistant cows sequenced with the Illumina NovaSeq 6000 platform in the paired-end mode with 150 bp reads length. In this group, the number of reads available per individual ranged between 311,675,740 and 908,861,126.

### 4.2. Genotyped Animals

Another set of Polish Holstein–Friesian cows with clinical mastitis records was obtained from the PLOWET database (accessed on 17 September2022) used for veterinarian-recorded health traits in four experimental dairy farms belonging to the National Institute of Animal Production. Among 1499 individuals, clinical mastitis was recorded for 712 cows. Most cows were genotyped using the Illumina BovineSNP50K bead chip version 2. Cows genotyped with other commercial platforms were imputed to the above chip using Flmpute (v.2.2) software [49]. Genotype preprocessing comprised removing SNPs with a minor allele frequency below 0.01 and call rate below 99%, which resulted in 53,557 SNPs remaining for downstream analysis. Furthermore, multigenerational pedigree records since 1914, comprising 8944 ancestors of genotyped cows, were available for the estimation of their additive genetic relationship.

### 4.3. Data Processing

The first part of the analysis, that is, the bioinformatic pipeline, aimed to estimate the set of SNPs that form the input for the DL-based classification scheme. Then, the statistical pipeline was used for the selection of the single best-classifying model comprising its underlying neural network architecture, hyperparameters, and the subset of SNPs and cut-off values estimations. Finally, the biological pipeline was imposed on significant SNPs from the best classifying model to provide the relevant biological explanation of the data set consisting of genome annotation and enrichment analysis. All study protocols are visualized in Figure 7.

### 4.4. Bioinformatic Pipeline

The bioinformatics pipeline for SNP identification consisted of (i) the quality control step performed using the FASTQC software (v0.11.7) [50], (ii) the quality-based raw data filtering step with Trimmomatic (v.039) [51], (iii) alignment to the ARS-UCD1.2 reference genome (NCBI accession number: PRJNA391427) with BWA-MEM (v0.7.17-r1188) [52], (iv) the post-alignment processing step with the Samtools (v1.2) [53] and Bedtools (v2.21.0) [54] packages, (v) the SNP call using the GATK package (v4.1.9.0) [55], and (vi) the SNP filtering step with VCFtools (v 0.1.12b) [56]. In detail, based on the quality control report (i), the decision on trimming low-quality sequences was made. The procedure was carried out by scanning each read with a sliding window of 4 bases and trimming it when the average of the 4-base qualities fell below 20. The minimum read length after trimming was set to 60 bp. The alignment of short reads to the reference genome was performed with default parameters. Standard post-alignment processes included sorting and indexing aligned sequences, removing PCR duplicates, and further quality control. The variant calling step followed the best-practice protocol provided by van der Auwera and O’Connor [57]. Variant filtering included removing variants with more than one alternative allele, identification quality below 20, and read depth at the variant site below 10. After all the above-mentioned edits, cows with average genome coverage below 7 were removed from the downstream analyses.

### 4.5. Statistical Pipeline

#### 4.5.1. Logistic LASSO Regression

The first step to overcome the p >> n problem, SNP preselection was performed by applying a logistic regression model:(1)$$P\left(y=1|X\right)=\frac{e^{\beta^{T}X}}{1+e^{\beta^{T}X}},$$
where $P\left(y=1|X\right)$ represents the probability of being mastitis-susceptible for each cow conditional on SNP genotypes $\left(X\right)$ with a LASSO [58] penalty $\left(\lambda \right)$ imposed on the SNP effect estimator ($\hat{\beta }$):(2)$$\hat{\beta }=\arg{min}_{\beta }\left\{-\sum_{i=1}^{N_{cow}}\left[y_{i}log\left(\beta^{T}x_{i}\right)+\left(1-y_{i}\right)log\left(1-\beta^{T}x_{i}\right)\right]+\lambda \sum_{j=1}^{N_{SNP}}\left|\beta_{j}\right|\right\}.$$

This logistic regression model was implemented with Python through the Scikit learn library [59] using the incremental gradient likelihood optimization method with support for non-strongly convex composite objectives [60] and the L1 regularization for SNP effect estimation. The penalty was expressed as $C=\frac{1}{\lambda }$, while various penalties were implemented using a grid on C within the interval (0.1; 1.0] with a step of 0.1. Note that the smaller the λ, the more SNP estimates will be set to zero.

#### 4.5.2. The Deep Learning Algorithm

SNP sets preselected by LASSO with different penalties were also used in a deep learning classifier that was implemented through the Keras interface (https://keras.io/) with the TensorFlow [61] library in Python. The rectified linear unit (ReLU) function: $\left(u\right)\;=\;\max\left(0,\;u\right)$, was used as the activation function for all, except the last layer, for which the sigmoid function was applied: $g\left(u\right)=\frac{1}{1+e^{-u}}$ where u is the node value calculated using the total sum of the input node values assigned to them. After each layer, dropout regularisation was applied, resulting in a predefined fraction of the input units being set to zero. The Adam algorithm [62] which implements the stochastic gradient descent approach was used to optimize the binary loss function of cross-entropy.

#### 4.5.3. Hyperparameter Tuning and Validation

During the learning process, no fixed DL architecture was imposed, instead, the final architecture comprising the number of layers and neurons per layer, the dropout rate within each layer, and the learning rate for the optimization algorithm was selected from the set of architectures dynamically sampled using the Optuna (v3.6) software [63] with one fixed hyperparameter label smoothing, which transformed binary labels into probabilities. In particular, the TPESampler implemented in the Optuna software was used for searching over DL algorithm hyperparameters with the number of iterations, i.e., a single execution of hyperparameter estimation was set to 50. The predefined range of sampled hyperparameters was given (Table 2), which additionally summarizes parameters that were set to fixed values, i.e., were not estimated, by the Optuna software. A sampling of DL architectures was performed separately for each SNP subset defined by different LASSO penalties $\left(C\right)$. Additionally, the following mechanisms were implemented to prevent overfitting: (i) the early stopping mechanism, which terminates training of a given DL architecture when a value of the loss function did not decrease for 5 epochs; (ii) the pruning algorithm based on AUC, implemented via the MedianPruner method that terminates learning for DL architectures that result in small AUC. To avoid overfitting, the training process was evaluated using a four-fold cross-validation.

#### 4.5.4. The Estimation of the Optimal Cut-Off Point

The final DL architecture estimated for each LASSO penalty was applied to classify the test data set. For each individual, the output of the last layer, resulting from the sigmoid activation function, was expressed as the probability of being mastitis-susceptible. However, instead of applying a default 0.5 cut-off, for each of the selected DL architectures, the optimal probability cut-off was estimated using the cutpointR package (v1.1.2) [64] implemented in R, based on the classification of cows from the training data set. In particular, the algorithm implemented into cutpointR determines the optimal cut-off value by maximizing the ACC metric. Estimates of the optimal cut-off values were obtained based on 1000 bootstrap samples of the training data set.

#### 4.5.5. The Selection of Significant SNPs

For each SNP, SHAP values were used to assess the importance of each SNP on the classification:(3)$$SHAP_{ij}\left(i\right)=\sum_{S\subseteq F\backslash \{j\}}^{\;}\frac{\left|S\right|!\left(\left|F\right|-\left|S\right|-1\right)!}{\left|F\right|!}{[P}_{F}\left(i\right)-P_{S}\left(i\right)].$$
where F denotes a full set of SNPs, S is a subset of F with j-th SNP removed, $P_{F}\left(i\right)$ represents a probability of an i-th individual being mastitis-susceptible estimated based on full SNP set F, and $P_{S}\left(i\right)$ represents a probability of an i-th individual being mastitis-susceptible estimated based on the subset S. Due to a very large number of SNPs, SHAP values were not calculated directly, following the above formula, but were computed approximately using the DeepExplainer (v0.45.0) [65]. The SHAP values were then rescaled to z scores: $z_{i}=\frac{\bar{{SHAP}_{j}}-\hat{\mu }}{\hat{\sigma }}$, where $\bar{{SHAP}_{j}}$ represents the mean of the SHAP values calculated for the j-th SNP across all individuals, $\hat{\mu }$ represents the mean and $\hat{\sigma }$ is the standard deviation of $\sum_{j=1}^{N_{SNP}}E\left({SHAP}_{j}\right),$ to assess the significance of each SNP’s significance by testing $\left(H0:z_{i}\leq 0\;vs.H1:z_{i}>0\right)$ based on *p*-values from the standard normal distribution. Each *p*-value was transformed to a false discovery rate (FDR) [66] to account for multiple testing.

#### 4.5.6. The Evaluation of DL Classifiers

The AUC metric was used as an indicator of the performance of each DL [67] while for the selection of the final, that is, the best classifier, sensitivity (SENS), specificity (SPEC), accuracy (ACC), and Matthews correlation coefficient (MCC) metrics were computed. These metrics are based on the following classification outcomes:

True positive (TP), defined as the scenario in which a mastitis-susceptible individual was classified as mastitis-susceptible.False positive (FP), defined as the scenario in which a mastitis-resistant individual was classified as mastitis-susceptible.True negative (TN), defined as the scenario in which a mastitis-resistant individual was classified as mastitis-resistant.False negative (FN), defined as the scenario in which a mastitis-susceptible individual was classified as mastitis-resistant.

They were defined as $ACC=\frac{TP\;+\;FN}{TP\;+\;TN\;+FP\;+\;FN},SENS=\frac{TP}{TP+FN},SPEC=\frac{TN}{FP+TN},MCC=\frac{TP\cdot TN-FP\cdot FN}{\sqrt{(TP+FP)\cdot \left(TP+FN\right)\cdot \left(TN+FP\right)\cdot \left(TN+FN\right)}}$. Among DL architectures with large AUC with a corresponding 95% confidence interval [68], the best classifier was then chosen based on the highest accuracy, sensitivity, and specificity.

### 4.6. Biological Pipeline

SNPs from the best algorithm with FDR below 0.05 were annotated to genes from the ARS-UCD1.2 reference assembly by the Variant Effect Predictor (VEP) [69] considering the maximum distance upstream/downstream of 5000 bp from the closest gene. Furthermore, for genes marked by significant SNPs, enrichment analysis was performed using the Database for Annotation, Visualization, and Integrated Discovery (DAVID v2021) tool [70] using the most specific levels of Gene Ontologies [71] defined for biological processes, molecular functions, and cellular components, as well as the metabolic pathways defined by the databases KEGG [72] and Reactome [73].

### 4.7. Genome-Wide Association Study for Clinical Mastitis in Genotyped Cows

The association study was carried out using a multi-SNP approach based on the following mixed linear model
(4)$$y=X\beta +Z_{1}q+Z_{2}a+\varepsilon ,$$
where y is a binary clinical mastitis status, β is a vector of fixed effect represented by a general mean and age at diagnosis, vector q contains SNP effects, a is a vector of additive polygenic effects of cows that were not explained by SNP genotypic variation, and vector ε contains error terms. $Z_{1}$ is a design matrix for SNP genotypes, which was parameterized as −1, 0, or 1 for a homozygous, heterozygous, and an alternative homozygous SNP genotype, respectively. The covariance structure of the model is given as follows.

$q\sim N\left(0,I\frac{{\hat{\sigma }}_{a}^{2}}{N_{snp}}\right)$, with I being an identity matrix, ${\hat{\sigma }}_{a}^{2}=0.06\;{\hat{\sigma }}_{y}^{2}$ representing the additive genetic variance component, and $N_{snp}$ being equal to the number of SNPs (53,557);$a\sim N\left(0,A{\hat{\sigma }}_{a*}^{2}\right)$, where A is the numerator relationship matrix calculated based on the pedigree relationship and ${\hat{\sigma }}_{a*}^{2}$ is the rest of additive genetic variance that was not explained by SNPs ${0.05\;\hat{\sigma }}_{a}^{2}$;$\varepsilon \sim N\left(0,I{\hat{\sigma }}_{\varepsilon }^{2}\right)$ where I is an identity matrix and ${\hat{\sigma }}_{\varepsilon }^{2}=0.92\;{\hat{\sigma }}_{y}^{2}$ representing the residual variance.

The estimation of model effects was based on solving the mixed model equations introduced by Henderson [74]:(5)$$\left[\begin{array}{c} \hat{\beta }\\ \hat{q}\\ \hat{a} \end{array}\right]={\left[\begin{array}{ccc} X^{T}R^{-1}X & X^{T}R^{-1}Z_{1} & X^{T}R^{-1}Z_{2}\\ Z_{1}^{T}R^{-1}X & Z_{1}^{T}R^{-1}Z_{1}+G_{1}^{-1} & Z_{1}^{T}R^{-1}Z_{2}\\ Z_{2}^{T}R^{-1}X & Z_{2}^{T}R^{-1}Z_{1} & Z_{2}^{T}R^{-1}Z_{2}+G_{2}^{-1} \end{array}\right]}^{-1}\left[\begin{array}{c} X^{T}R^{-1}y\\ Z_{1}^{T}R^{-1}y\\ Z_{2}^{T}R^{-1}y \end{array}\right]$$
where $R=I{\hat{\sigma }}_{\varepsilon }^{2}$, $G_{1}=I\frac{{\hat{\sigma }}_{a}^{2}}{N_{snp}}$and $G_{2}=A{\hat{\sigma }}_{a*}^{2}$. Consequently, the variance of y is given by $Z_{1}G_{1}Z_{1}^{T}+Z_{2}G_{2}Z_{2}^{T}+R$. Note that the variance components (${\hat{\sigma }}_{a}^{2}$ and ${\hat{\sigma }}_{\varepsilon }^{2}$) were not estimated in this study, but were assumed as known based on the parameters estimated elsewhere (unpublished internal evaluation) using a larger cohort.

For testing the hypotheses ($H_{0}:q=0$ vs. $H_{1}:q\neq 0$), we used the Wald test: $W=\frac{\hat{q}}{\sigma_{\hat{q}}}$, where $\sigma_{\hat{q}}$ is the standard error of the estimated SNP effect $\hat{q}$. Under $H_{0}$ this statistic follows the standard normal distribution. The multiple testing correction was carried out via Bonferroni. The positions were remapped from UMD3.1 to the ARS-UCD1.2 reference genome using the NCBI Genome Remapping Service [75] with default settings (minimum base ratio for remapping = 0.5 and maximum difference ratio between source and target length = 2.0) and annotated to genes from the ARS-UCD1.2 reference assembly using the VEP tool.

## 5. Conclusions

With this contribution, we emphasize the importance of exploiting multiple aspects of the bioinformatic analysis of biological data, which go beyond the application of bioinformatic software and also comprise elements of feature selection in the multidimensional data that are nowadays typical in genomics, multilevel model selection, statistical analysis, and finally biological interpretation. This approach is especially important in the analysis of phenotypes with a complex mode of inheritance, such as, in our case, clinical mastitis, which is influenced by multiple genes of varying effects and not necessarily by just a few genes of large effects. Furthermore, due to the low to moderate heritability, the effect of these genes is likely also dependent on the environment (although this concept was not formally tested in our study).

In addition, we demonstrated that, since, in biology, due to financial, ethical, or data availability constraints, it is not always possible to obtain very large data sets, machine learning applications need to carefully focus on selecting models’ architectures and their hyperparameters. In the case of a limited size of input data, only such an extensive model selection approach allows reasonable classification accuracy (as in our case) or accurate prediction to be obtained.

## Figures and Tables

**Figure 1 ijms-25-04715-f001:**
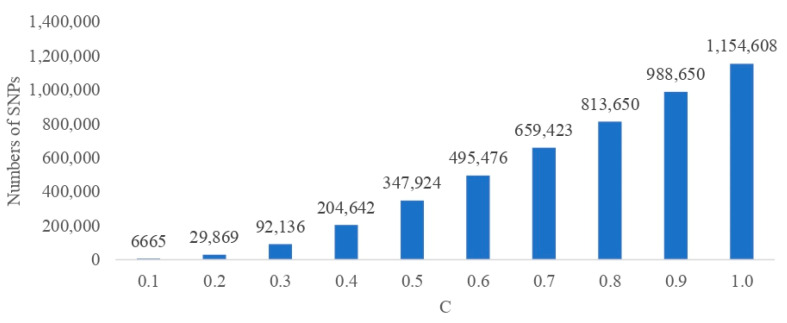
Numbers of Single Nucleotide Polymorphisms (SNPs) selected using the LASSO logistic regression for each model with different penalties (C).

**Figure 2 ijms-25-04715-f002:**
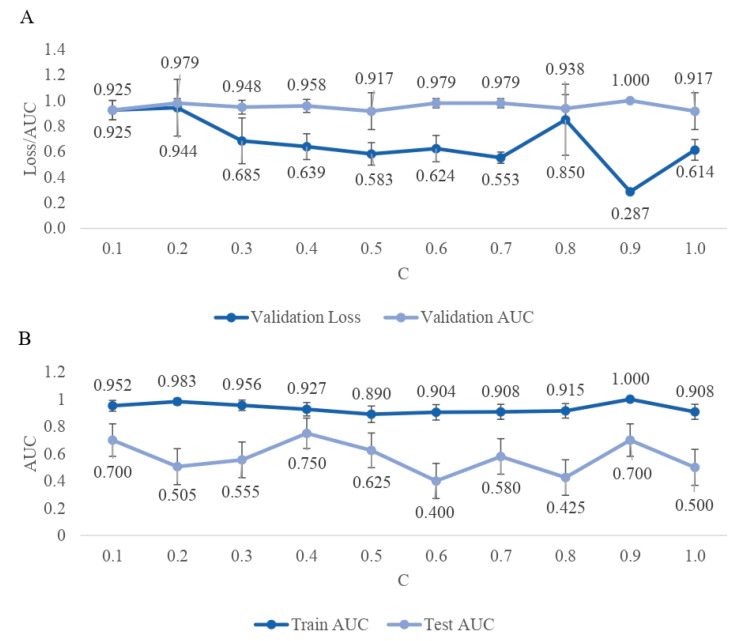
(**A**). The Area Under the Curve (AUC) and Loss based on the 4-fold cross-validation of the training data set. (**B**). AUC calculated for training and test data.

**Figure 3 ijms-25-04715-f003:**
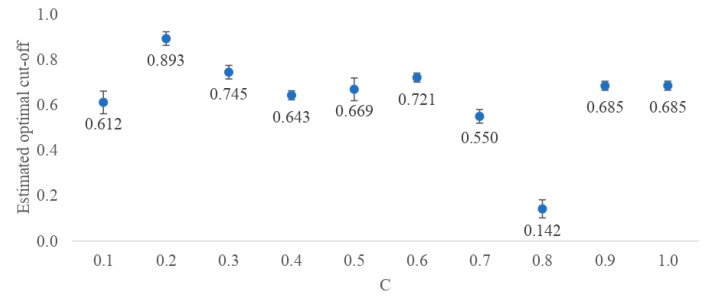
Probability cut-off values for mastitis classification into the susceptible or resistant group estimated based on the optimization for accuracy metric, for models with different penalties (C). The standard deviations for each estimate were calculated using the out-of-the-bag samples.

**Figure 4 ijms-25-04715-f004:**
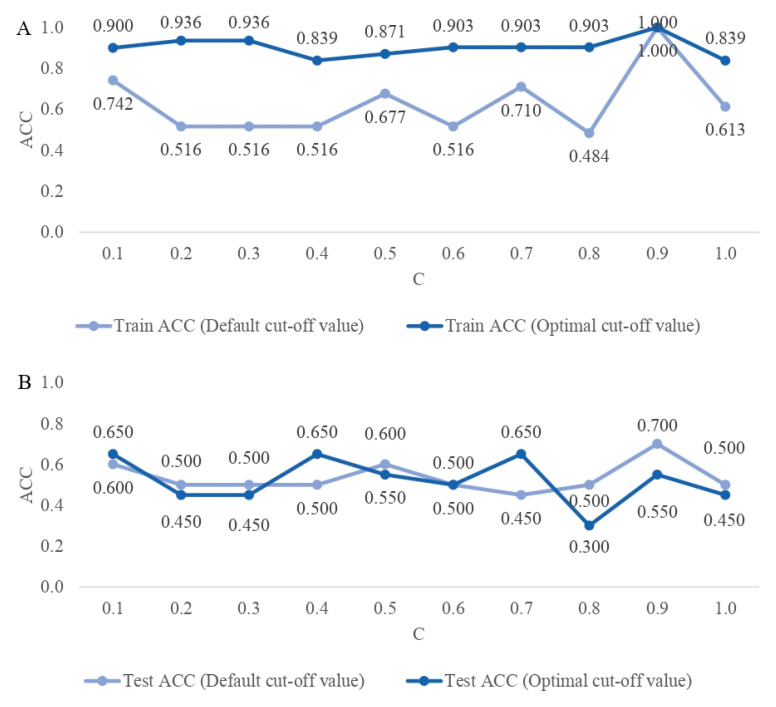
(**A**). The classification accuracy (ACC) for the training data set resulting from using the 0.5 cut-off (default) and the estimated cut-off (optimal) values, for models with different penalties (C). (**B**). ACC for the test data set resulting from using the 0.5 cut-off (default) and the estimated cut-off (optimal) values, for models with different penalties (C).

**Figure 5 ijms-25-04715-f005:**
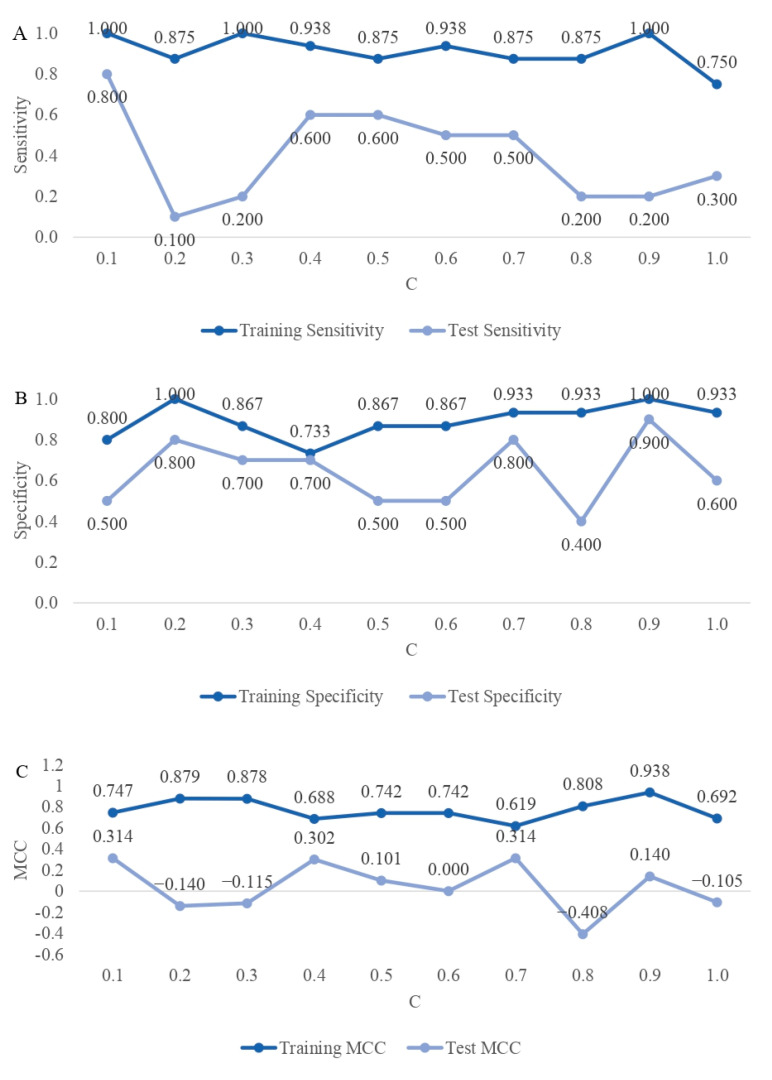
(**A**). Classification sensitivity of training and test data sets based on the optimal cut-off values. (**B**). Classification specificity of training and test data sets based on the optimal cut-off values. (**C**). Classification Matthews correlation coefficients (MCC) of training and test data sets based on the optimal cut-off values. Results shown for models with LASSO different penalties (C).

**Figure 6 ijms-25-04715-f006:**
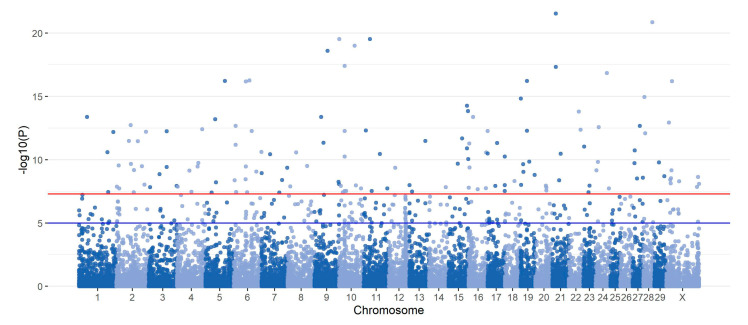
Manhattan plot with absolute values of $\bar{{SHAP}_{j}}$ for the best classifying model. The red horizontal line represents the genome-wide significance threshold of *p*-value = $5.0\times 10^{-8}$ and the blue horizontal line represents the suggestive significance threshold of *p*-value = $1.0\times 10^{-5}$.

**Figure 7 ijms-25-04715-f007:**
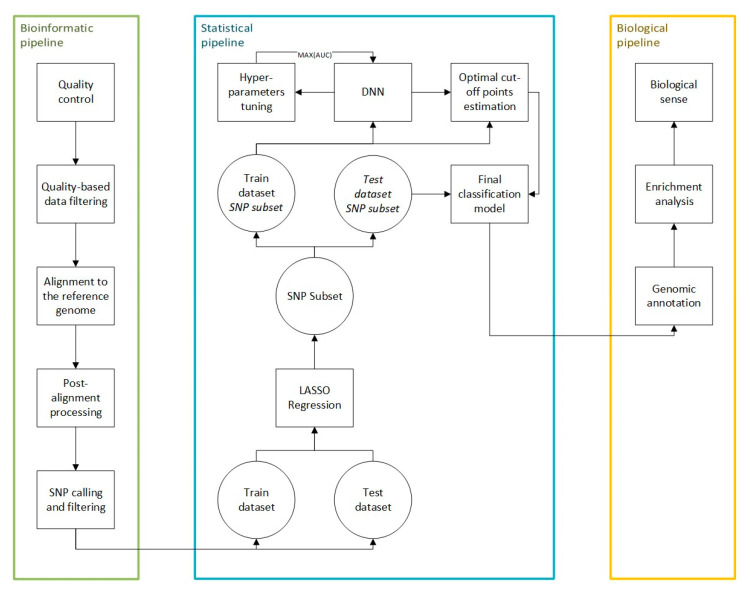
Flow diagram of data analysis.

**Table 1 ijms-25-04715-t001:** The optimal DL architecture estimated for each SNP set.

$C=\frac{1}{\lambda }$	N Layers	N Units Inside Each Layer	Dropout Rate within Each Layer	Learning Rate
0.1	1	[31]	[0.285]	3.026 × 10^−09^
0.2	3	[32; 36; 37]	[0.302; 0.218; 0.311]	4.263 × 10^−10^
0.3	3	[11; 16; 12]	[0.323; 0.242; 0.243]	7.147 × 10^−09^
0.4	2	[7; 46]	[0.210; 0.358]	2.328 × 10^−11^
0.5	2	[48; 45]	[0.312; 0.250]	6.700 × 10^−12^
0.6	3	[47; 37; 28]	[0.398; 0.278, 0.300]	6.900 × 10^−09^
0.7	2	[10; 18]	[0.215; 0.222]	7.896 × 10^−09^
0.8	4	[35; 35; 26; 13]	[0.323; 0.261, 0.257; 0.327]	4.268 × 10^−10^
0.9	1	[50]	[0.250]	6.829 × 10^−09^
1.0	3	[23; 49; 9]	[0.297; 0.362, 0.365]	1.698 × 10^−09^

**Table 2 ijms-25-04715-t002:** Hyperparameters of the DL algorithm sampled by the Optuna software or treated as fixed.

Sampled Hyperparameters	Range
Number of layers	[1, 6]
Number of units per layer	[4, 50]
Dropout rate	[0.2, 0.4]
Learning rate	[1.0 × 10^−12^, 1.0 × 10^−8^]
**Fixed hyperparameters**	
Number of epochs	300
Label smoothing	0.2

## Data Availability

The deoxyribonucleic acid sequences of the 32 and 20 cows from the training data set are available from the NCBI BioProject database under the accession IDs PRJNA359667 and PRJNA979229, respectively.

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
