# Peer review of "An Explainable Deep Learning Classifier of Bovine Mastitis Based on Whole-Genome Sequence Data—Circumventing the p >> n Problem"

_ijms, 2024, doi:10.3390/ijms25094715_

Round 1

Reviewer 1 Report

Comments and Suggestions for Authors

Rewiew Manuscript ID: ijms-2965216

Brief summary: The authors start from the correct consideration that one of the main problems of dairy farms is "mastitis disease", which in fact causes, among other things, milk production losses, administration of drugs, especially antibiotics, and animal (cows) reform. In the paper entitled "An explainable deep learning classifier of bovine mastitis based on whole genome sequence data - circumventing the p>>n problem", the authors approach the problem by combining LASSO logistic regression with deep learning to classify cows as susceptible or resistant to mastitis based on single nucleotide polymorphism (SNP) genotypes. Among many architectures, the one with 204,642 SNPs was selected as the best, it was composed of 2 layers with 7 and 46 units per layer, respectively, implementing dropout rates of 0.210 and 0.358, the classification of the test data gave the result Area Under Curve=0.750, Accuracy=0.650, Sensitivity=0.600 and Specificity=0.700. The authors conclude that the optimal approach can predict mastitis susceptibility or resistance status in approximately 65% of cows. Genes marked by the most significant SNPs are related to immune response and protein synthesis.

General remarks: Before presenting my thoughts on the article, it is necessary to make a premise: dairy animals, especially dairy cows, can manifest mastitis phenomena at different times during their 9-10 month lactation. This pathology may manifest itself with clinical symptoms visible to the farmer (e.g. traces of blood in the milk, swollen quarters, etc.) or it may not be visible to the naked eye (e.g. subclinical mastitis) but can be diagnosed by laboratory tests (e.g. increase in somatic cell count above physiological levels). Authors should contextualise these important concepts in the materials, methods and discussion.

Comments

This paper is in line with the topic of the Journal, but in its current form has some minor limitations, which I list below.

Introduction: comprehensive, with appropriate bibliographical references, I suggest arguing the purpose of the study better.

Results and discussion: sufficiently described, authors need to contextualise the "General remarks" above.

lines 104-105: table 1, rewrite "learning note" in the correct form ex 3.026 e-09

lines 287-297: another mastitogenic micro-organism (as contagious as Staph. aureus) is Streptococcus agalactiae, which must be discussed.

Materials and methods: some points need to be improved, in particular the methodology used to detect mastitic disease (clinical, subclinical, bacteriological tests, etc.).

Fig. 7 needs to be improved.

Conclusions: adequate.

References: should be given in MDPI style, e.g. year in bold.

Author Response

General remarks: Before presenting my thoughts on the article, it is necessary to make a premise: dairy animals, especially dairy cows, can manifest mastitis phenomena at different times during their 9-10 month lactation. This pathology may manifest itself with clinical symptoms visible to the farmer (e.g. traces of blood in the milk, swollen quarters, etc.) or it may not be visible to the naked eye (e.g. subclinical mastitis) but can be diagnosed by laboratory tests (e.g. increase in somatic cell count above physiological levels). Authors should contextualise these important concepts in the materials, methods and discussion.

AU: The materials and discussion were extended to incorporate the definition of mastitis diagnosis.

Introduction: comprehensive, with appropriate bibliographical references, I suggest arguing the purpose of the study better.

AU: The introduction was modified according to the Reviewer’s suggestions.

Results and discussion: sufficiently described, authors need to contextualise the "General remarks" above.

AU: The discussion was modified.

lines 104-105: table 1, rewrite "learning note" in the correct form ex 3.026 e-09

AU: The notation was modified.

lines 287-297: another mastitogenic micro-organism (as contagious as Staph. aureus) is Streptococcus agalactiae, which must be discussed.

AU: The Streptococcus agalactiae species was included in the discussion section.

Materials and methods: some points need to be improved, in particular the methodology used to detect mastitic disease (clinical, subclinical, bacteriological tests, etc.).

AU: The information about the mastitis diagnosis was included in the methodology section.

Fig. 7 needs to be improved.

AU: The Figure 7 was modified and the quality of all figures were improved.

References: should be given in MDPI style, e.g. year in bold.

AU: The references were modified.

Reviewer 2 Report

Comments and Suggestions for Authors

In this paper, the authors present an approach that combines LASSO logistic regression with deep learning to address the p>>n problem in the biological annotation of whole-genome sequence data, aiming to classify cows as mastitis-susceptible or mastitis-resistant based on SNPs. However, the following lists some comments. Firstly, the details of the combined method are not fully explained, making it difficult for readers to understand its intricacies, especially for the large p small n problem. Secondly, the evaluation of the model's performance lacks comprehensive analysis, such as discussing its performance across different categories or conducting cross-validation.  The selected SNPs are also related to the LASSO panelty. Furthermore, the analysis of significant SNPs and GO terms is not sufficiently deep, lacking an exploration of their specific functions and potential applications.

Comments on the Quality of English Language

English need be polished before publication.

Author Response

Firstly, the details of the combined method are not fully explained, making it difficult for readers to understand its intricacies, especially for the large p small n problem. 

AU: The method section was modified according to the Reviewer’s suggestion.

Secondly, the evaluation of the model's performance lacks comprehensive analysis, such as discussing its performance across different categories or conducting cross-validation. 

AU: The 4-cross-validation was conducted in the pipeline and described in the original version of the manuscript. In the revised version, the cross-validation results were emphasised in the method section.

The selected SNPs are also related to the LASSO panelty. Furthermore, the analysis of significant SNPs and GO terms is not sufficiently deep, lacking an exploration of their specific functions and potential applications.

AU: The role of GO enrichment analysis for genes affected by significant SNPS was emphasized.

Comments on the Quality of English Language: English need be polished before publication.

AU: The language was improved.

Round 2

Reviewer 2 Report

Comments and Suggestions for Authors

My comments have been addressed.

Comments on the Quality of English Language

English need be polished before publication.